# An Energy Perspective in Knowledge Distillation

## Abstract

Knowledge distillation is a widely studied technique for transferring knowledge from a large teacher model to a smaller student model, with the aim of maintaining high performance while reducing computational complexity. However, the performance of the student model often suffers when the teacher model is overly large. We observe significant differences in the ability of teacher and student models to minimize losses, with student models exhibiting higher entropy. This underscores the inherent difficulty in transferring knowledge from the more complex teacher model to the simpler student model. Through theoretical analysis, we propose a straightforward intermediate alignment module to narrow the gap between the student and the teacher, thus enhancing the student performance. Compared with vanilla distillation, the proposed method has the potential to improve the performance of the student model when the teacher model is significantly large, paving the way for more efficient and powerful model learning techniques in the field of knowledge distillation.

## 1 Introduction

The field of deep learning has seen tremendous progress in recent years, with the development of large-scale neural networks that achieve state-of-the-art performance on a variety of tasks [6, 10, 38, 33, 13]. However, these models are often computationally expensive and memory-intensive, making them difficult to deploy on resource-constrained devices. Therefore, model compression has attracted considerable research attention in recent years [11, 26, 5, 14, 7, 34]. One of the predominant approaches in model compression is knowledge distillation (KD) [11, 30, 9, 29, 3], which trains a smaller model (i.e., the student) with the output of a larger and well-trained model (i.e., the teacher).

Naturally, one would expect to train a better student with a larger and more accurate teacher. However, recent research has invalidated this hypothesis and found that knowledge distillation suffers from a mysterious performance degradation problem [4, 20]. Specifically, the student performance degrades with an oversized teacher, indicating that the knowledge of an oversized teacher cannot be effectively transferred to the student (Table 1). This problem may be due to the fact that directly transferring the teacher output ignores the inherent differences between the teacher network and the student network, and previous method found that a intermediate translation layer can help to improve the student performance [14].

This paper investigates the problem from the perspective of the energy gap between the teacher and student models. For instance, if we train both the student and the teacher model with cross entropy loss, the student model may have difficulty in achieving a comparably low cross-entropy loss as the teacher model, therefore it is expected that the student model's entropy are higher than the teacher model. When we train the student to mimic the teacher output, the discrepancy in entropy may potentially hinder the student's ability to effectively learn from the teacher.

First, we define the energy gap via two metrics, entropy and free energy. Free energy has a close relationship with entropy [19] and can be expressed as the log partition function of softmax. Entropy gap $G_{entropy}$ is the difference between the student entropy and the teacher entropy, and free energy gap $G_f$ is defined similarly as follows, where p and q represent the student and the teacher probability,

z and v represent the student and the teacher output logits:

$$G_{entropy} = S(p) - S(q) = \sum_i -p_i \log(p_i) + \sum_i q_i \log(q_i)$$

$$G_f = F(v) - F(z) = \tau \log \sum_i e^{v_i/\tau} - \tau \log \sum_i e^{z_i/\tau} \tag{1}$$

Second, we introduce a simple intermediate alignment layer to narrow the energy gap between the student and the teacher. Through rigorous mathematical analysis, we find that the logits norm of the model is crucial in solving the problem. By employing a normalization layer that aligns the logits norm of the student model and the teacher model, the energy gap between the two can be minimized effectively. We term our method spherical knowledge distillation (SKD) because both the student and teacher logits have the same norm, therefore lie on the same hypersphere.

The experimental results demonstrate that our method is an effective approach to knowledge distillation, achieving excellent performance on mainstream tasks. Our method is computationally efficient and easy to implement, making it a practical solution for deploying deep learning models on resource-constrained devices.

## 2 METHODOLOGY

### 2.1 BACKGROUND

**Vanilla Knowledge Distillation**   When training neural networks, we minimize the negative log-likelihood of the ground truth class to update model parameters. After the model is well-trained, the probability of the ground truth would be close to 1, while the probabilities of other labels are near 0. Hinton et al. [11] noticed that the small wrong probabilities of large models are useful to unveil "dark knowledge". For example, given a picture of a "cat", the model is more likely to output a higher probability for class "dog" than class "airplane". These wrong probabilities imply the relationship between the two classes and unveil how a model tends to generalize. This observation inspired the usage of large models' outputs as soft targets to train efficient small models, which is the core idea of Knowledge Distillation. However, modern deep networks tend to produce peaky probabilities [8, 18], i.e., the numbers of those wrong classes (near zero values) would be negligible compared to the ground truth (near one). Thus Hinton et al. [11] proposed to raise the temperature of the last softmax layer to soften the output probabilities, which can be used as soft targets to train student networks.

For vanilla knowledge distillation, the KD loss can be defined as follows:

$$\mathcal{L}_{KD} = -\sum_i q_i \log p_i$$

$$p_i = \frac{e^{z_i/\tau}}{\sum_j e^{z_j/\tau}}, q_i = \frac{e^{v_i/\tau}}{\sum_j e^{v_j/\tau}} \tag{2}$$

Where we denote logits of the teacher as $v$, logits of the student as $z$, student probability as $p$, teacher probability as $q$, temperature as $\tau$, and the $i$-th and $j$-th value of logits (i.e. the $i$-th and $j$-th category of K classes) as $i$ and $j$, respectively. The final loss for the student is then the weighted sum of the typical cross entropy loss $\mathcal{L}_{cls}$ and the knowledge distillation loss $\mathcal{L}_{KD}$:

$$\mathcal{L} = \lambda \mathcal{L}_{KD} + (1 - \lambda)\mathcal{L}_{cls} \tag{3}$$

The popular choice of the temperature $\tau$ is in $\{3, 4, 5\}$, and the weight $\lambda$ is 0.9 [11, 4, 31].

### 2.2 PERFORMANCE DEGRADATION PROBLEM AND ENTROPY GAP

While knowledge distillation achieved success in many fields, a mysterious performance degradation problem was observed [4, 20]. Since the idea of knowledge distillation is transferring teacher knowledge to students, one natural hypothesis is that a larger and more accurate teacher would capture more knowledge and thus train better students. However previous studies invalidate this hypothesis by showing that student performance degenerates unexpectedly with larger teachers. For

Table 1: The performance degradation problem.

| Teacher | ResNet20 | ResNet32 | ResNet44 | ResNet56 | ResNet110 |
|---|---|---|---|---|---|
| Teacher Acc | 69.57 | 70.9 | 71.9 | 72.8 | **73.8** |
| Student Acc | 67.4 | **68.2** | 68 | 67.5 | 67.1 |
| KD loss | **1.1** | 1.7 | 2.1 | 2.5 | 3.3 |

example, as shown in Table 1, applying a larger teacher model will increase the KD loss and decrease the accuracy of the student (with ResNet14 as the backbone).

In order to address performance degradation issues, we introduce the concept of the energy gap. Taking entropy as an example, a larger-capacity network is more likely to have low entropy. This is because a larger-capacity network manages to further minimize the cross-entropy loss, even when it can correctly classify almost all training samples. In this situation, the student model may struggle to match the teacher output. It follows that, compared to larger teacher models, the student model might perform better when matching the outputs of smaller teacher models.

In the following subsections, we first analyze the gap using two energy-based metrics. Then we provide a theoretical analysis showing how to reduce the gap between the teacher and student to alleviate the performance degradation problem. When the gap is narrowed, we show that student could perform better than the vanilla knowledge distillation in experiments.

## 2.3 ANALYSIS VIA ENTROPY AND FREE ENERGY

**Entropy and free energy**   We apply two energy-based metrics termed *Entropy* and *Helmholtz Free Energy*, that are commonly used in quantifying thermal processes in thermodynamics.

The concept of *Entropy* was also introduced into information science [28] to measure the uncertainty of a random variable [16], and is defined as follows:

$$S = -\sum_i p_i \log(p_i) \tag{4}$$

*Helmholtz Free Energy* is frequently used in energy-based model [17]. Entropy and Helmholtz free energy has a close relationship with each other [19]. Helmholtz free energy can be expressed as the log partition function as follows, where $z_i$ is the $i$-th element of the logits, and $\tau$ is the temperature:

$$F(z) = \tau \log \sum_i e^{z_i/\tau} \tag{5}$$

It can be noted that when the temperature $\tau$ is set to 1, $F(z)$ is equivalent to the RealSoftMax $LSE(x) = \log \sum_i e^{x_i}$ [22]. Therefore, $F(z)$ can be regarded as a smooth approximation to the maximum function.

We measured the entropy and Helmholtz free energy for different models trained on the CIFAR-100 dataset with temperature $\tau = 1$ (with one-hot labels). As shown in Table 2, there exists a large gap in both entropy and free energy between networks with different sizes. And larger models generally produce lower entropy and higher free energy. In the following part of this subsection, we will first define *Entropy Gap* and *Helmholtz Free Energy Gap*, and then demonstrate how to reduce the gap. We use K to denote K classes and $||z||$ to denote the norm of z in the following analysis.

**Entropy Gap Analysis**   We define the entropy gap as follows:

$$G_{entropy} = S(p) - S(q) \tag{6}$$

**Theorem 1.** *Let p and q represent the student and the teacher output probability, z and v represent the student and teacher logits, and $\tau$ represents the temperature. The entropy gap $G_{entropy}$ approximate to $\frac{1}{K\tau^2}(||v||^2 - ||z||^2)$, under the assumption that z and v are zero meaned separately.*

Table 2: The gap of entropy and Helmholtz free energy with ResNet14 as the student.

|  | ResNet20 | ResNet32 | ResNet44 | ResNet56 | ResNet110 |
|---|---|---|---|---|---|
| Entropy | 0.74 | 0.45 | 0.35 | 0.22 | 0.09 |
| $G_{entropy}$ - Vanilla KD | 0.146 | 0.181 | 0.222 | 0.246 | 0.261 |
| $G_{entropy}$ - SKD | **0.053** | **0.074** | **0.081** | **0.083** | **0.094** |
| Free Energy | 13.11 | 13.84 | 14.45 | 15.37 | 16.13 |
| $G_f$ - Vanilla KD | 0.042 | 0.053 | 0.063 | 0.068 | 0.074 |
| $G_f$ - SKD | **0.029** | **0.042** | **0.041** | **0.045** | **0.044** |

*Proof.*

$$
\begin{aligned}
G_{entropy} &= S(p) - S(q) \\
&= \sum_i -p_i \log(p_i) + \sum_i q_i \log(q_i) \\
&= \sum_i (q_i v_i/\tau - p_i z_i/\tau) + \log \sum_j e^{z_j/\tau} - \log \sum_j e^{v_j/\tau} \qquad (7) \\
&= \sum_i \left( \frac{e^{v_i/\tau} v_i/\tau}{\sum_j e^{v_j/\tau}} - \frac{e^{z_i/\tau} z_i/\tau}{\sum_j e^{z_j/\tau}} \right) + \log \sum_j e^{z_j/\tau} - \log \sum_j e^{v_j/\tau}
\end{aligned}
$$

With Taylor expansion $e^x \approx 1 + x$:

$$
\begin{aligned}
G_{entropy} &\approx \sum_i \left( \frac{(1+v_i/\tau)v_i/\tau}{\sum_j (1+v_j/\tau)} - \frac{(1+z_i/\tau)z_i/\tau}{\sum_j (1+z_j/\tau)} \right) \\
&\quad + \log \sum_j (1+z_j/\tau) - \log \sum_j (1+v_j/\tau)
\end{aligned} \qquad (8)
$$

We follow the assumption from Hinton [11], that the logits have been zero-meaned separately for each training example so that $\sum_j z_j = \sum_j v_j = 0$.

Given the above assumption, we can get:

$$
\begin{aligned}
G_{entropy} &\approx \sum_i \left( \frac{(v_i/\tau)^2}{K} - \frac{(z_i/\tau)^2}{K} \right) + \log(K) - \log(K) \\
&= \frac{1}{K\tau^2} (||v||^2 - ||z||^2)
\end{aligned} \qquad (9)
$$

$\square$

**Helmholtz Free Energy Gap Analysis**  We define the Helmholtz free energy gap as follows:

$$
G_f = F(v) - F(z) \qquad (10)
$$

**Theorem 2.** *Let z and v represent the student and the teacher output logits. The Helmholtz free energy gap $G_f$ approximate to $\frac{1}{2K\tau}(||v||^2 - ||z||^2)$, under the assumption that z and v are zero meaned separately and $2\tau^2 K$ is large compared with the square of logits norm.*

*Proof.*

$$
\begin{aligned}
G_f &= F(v) - F(z) \\
&= \tau \log \sum_i e^{v_i/\tau} - \tau \log \sum_i e^{z_i/\tau} \\
&\approx \tau \log \left( K + \sum_i v_i/\tau + \frac{1}{2} \sum_i v_i^2/\tau^2 \right) \\
&\quad - \tau \log \left( K + \sum_i z_i/\tau + \frac{1}{2} \sum_i z_i^2/\tau^2 \right)
\end{aligned} \qquad (11)
$$

With Taylor expansion $e^x \approx 1 + x + \frac{1}{2}x^2$:

$$
\begin{aligned}
G_f &\approx \tau \log(K + \frac{1}{2}\sum_i v_i^2/\tau^2) - \tau \log(K + \frac{1}{2}\sum_i z_i^2/\tau^2) \\
&\approx \tau \log K(1 + \frac{1}{2\tau^2 K}||v||^2) - \tau \log K(1 + \frac{1}{2\tau^2 K}||z||^2) \\
&\approx \tau \log(1 + \frac{1}{2\tau^2 K}||v||^2) - \tau \log(1 + \frac{1}{2\tau^2 K}||z||^2) + \log K - \log K \\
&\approx \tau \log(1 + \frac{1}{2\tau^2 K}||v||^2) - \tau \log(1 + \frac{1}{2\tau^2 K}||z||^2)
\end{aligned}
\tag{12}
$$

When $2\tau^2 K$ is large compared with the square of logits norm, with $\log(1+x) \approx x$ for $|x| < 1$:

$$
G_f \approx \frac{1}{2K\tau}(||v||^2 - ||z||^2)
\tag{13}
$$

$\square$

## 2.4 SPHERICAL KNOWLEDGE DISTILLATION

**Alignment Layer** This module aligns the entropy and free energy levels of the student and the teacher by applying a mapping function on the logits. This mapping function is designed to minimizing the gap while preserve the inter-category knowledge in the teacher logits. Mathematically, let $S(x)$ and $F(x)$ denote entropy and the free energy function, z denote the student logits and v denote the teacher logits, the module seeks a transformation function $\mathcal{F}$ such that:

$$
S(\mathcal{F}(z)) \approx S(\mathcal{F}(v))
$$

$$
F(\mathcal{F}(z)) \approx F(\mathcal{F}(v))
$$

We have demonstrated that both Entropy Gap and Helmholtz Free Energy Gap can be represented in similar forms in the above section:

$$
\begin{aligned}
G_{entropy} &\approx \frac{1}{K\tau^2}(||v||^2 - ||z||^2) \\
G_f &\approx \frac{1}{2K\tau}(||v||^2 - ||z||^2)
\end{aligned}
\tag{14}
$$

It can be seen that $||v||^2 - ||z||^2$ plays a key role and can be removed by normalization techniques. Therefore, we propose Spherical Knowledge Distillation (SKD). Specifically, the alignment module is designed as a normalization operation, that student and teacher logits for each sample would be transformed as follows:

$$
\begin{aligned}
\mathcal{F}(z) &= \frac{z}{||z||} \\
\mathcal{F}(v) &= \frac{v}{||v||}
\end{aligned}
\tag{15}
$$

Let $\hat{z}$ and $\hat{v}$ denote the z and v after the transformation, the result would be $||\hat{v}||^2 = ||\hat{z}||^2$, which significantly reduce the gap for both entropy and Helmholtz free energy. The rest of SKD follows the standard distillation procedure:

$$
\begin{aligned}
p_i &= \frac{e^{\hat{z}_i/\tau}}{\sum_j e^{\hat{z}_j/\tau}}, q_i = \frac{e^{\hat{v}_i/\tau}}{\sum_j e^{\hat{v}_j/\tau}} \\
\mathcal{L}_{SKD} &= -\sum_i q_i \log p_i \\
\mathcal{L} &= \lambda \mathcal{L}_{SKD} + (1-\lambda)\mathcal{L}_{cls}
\end{aligned}
\tag{16}
$$

Table 3: CIFAR-100 experiments.

| Teacher
Student | WRN-40-2
WRN-16-2 | WRN-40-2
WRN-40-1 | ResNet56
ResNet20 | ResNet110
ResNet20 | ResNet110
ResNet32 | ResNet32*4
ResNet8*4 | VGG13
VGG8 |
|---|---|---|---|---|---|---|---|
| Teacher | 75.61 | 75.61 | 72.34 | 74.31 | 74.31 | 79.42 | 74.64 |
| Student | 73.26 | 71.98 | 69.06 | 69.06 | 71.14 | 72.50 | 70.36 |
| KD | 74.92 | 73.54 | 70.66 | 70.67 | 73.08 | 73.33 | 72.98 |
| FitNet | 73.58 | 72.24 | 69.21 | 68.99 | 71.06 | 73.50 | 71.02 |
| AT | 74.08 | 72.77 | 70.55 | 70.22 | 72.31 | 73.44 | 71.43 |
| SP | 73.83 | 72.43 | 69.67 | 70.04 | 72.69 | 72.94 | 72.68 |
| CC | 73.56 | 72.21 | 69.63 | 69.48 | 71.48 | 72.97 | 70.71 |
| VID | 74.11 | 73.30 | 70.38 | 70.16 | 72.61 | 73.09 | 71.23 |
| RKD | 73.35 | 72.22 | 69.61 | 69.25 | 71.82 | 71.90 | 71.48 |
| PKT | 74.54 | 73.45 | 70.34 | 70.25 | 72.61 | 73.64 | 72.88 |
| AB | 72.50 | 72.38 | 69.47 | 69.53 | 70.98 | 73.17 | 70.94 |
| FT | 73.25 | 71.59 | 69.84 | 70.22 | 72.37 | 72.86 | 70.58 |
| FSP | 72.91 | - | 69.95 | 70.11 | 71.89 | 72.62 | 70.23 |
| NST | 73.68 | 72.24 | 69.60 | 69.53 | 71.96 | 73.30 | 71.53 |
| CRD | 75.48 | 74.14 | 71.16 | 71.46 | 73.48 | 75.51 | 73.94 |
| **SKD** | **75.75** | **75.06** | **72.08** | **72.12** | **74.09** | **76.40** | **74.17** |

## 3 EXPERIMENTS

In this section, we show comprehensive experimental results to validate the effectiveness of SKD from several perspectives. Specifically, we first conducted experiments on two popular CV datasets to demonstrate the performance of SKD. Then we focused on evaluating whether SKD could alleviate the performance degradation problem.

**Dataset** 1) *CIFAR-100* [15] is a relatively small data set and is widely used for testing various deep learning methods. CIFAR-100 contains 50,000 images in the training set and 10,000 images in the evaluation set, with 100 fine-grained categories. 2) *ImageNet* [6] is a much larger dataset than CIFAR-100. ImageNet contains 1.2M images for training and 50K for validation, with 1,000 fine-grained categories. We used SGD with nesterov momentum 0.9, initial learning rate 0.1, batch size is set 256, weight decay $1 * 10^{-4}$ and temperature is set 4. Learning rate is dropped at 30, 60, 80 and 90 epoch.

**CIFAR Experimental settings** We ran a total of 240 epochs for all methods. The learning rate was initialized as 0.05, then it decayed by 0.1 every 30 epochs after 150 epochs. For MobileNetV2, ShuffleNetV1 and ShuffleNetV2, we use a learning rate of 0.01 as this learning rate is optimal for these models in a grid search, while 0.05 is optimal for other models. For both vanilla KD and SKD, we set the temperature as 4, weight as 0.9, and cross-entropy as 0.1 for all settings.

**ImageNet Experimental settings** ResNet18 was used as the student for all methods. The teacher network had been trained in advance of the experiments and was fixed during training. we use a learning rate of 0.1, total 100 epoches, and learning rate dropped by 0.1 in 30, 60, 80, 90 epoch, temperate is set to 4.

### 3.1 MAIN RESULTS

**Baselines** We selected various SOTA KD methods to evaluate the performances of SKD: FitNet [27], AT [39], SP [32], PKT [24], FT [14], FSP [37], CC [25], VID [1], CRD [31], RKD [23], KD [11], NST [12], ES [4], TA [20], SRRL [36]

**CIFAR-100** Table 3 shows that SKD always has higher accuracy than all other methods. In some situations (e.g. those where teacher/student ResNet110/ResNet32), the performances of SKD were even very close to those of the teacher.

Table 4: CIFAR-100 experiments when the teacher's architecture is significantly different.

| Teacher
Student | ResNet50
MoblieNetV2 | ResNet50
vgg8 | ResNet32*4
ShuffleNetV1 | ResNet32*4
ShuffleNetV2 | WRN-40-2
ShuffleNetV1 |
|---|---|---|---|---|---|
| Teacher | 79.34 | 79.34 | 79.42 | 79.42 | 75.81 |
| Student | 64.60 | 70.36 | 70.50 | 71.82 | 70.50 |
| KD | 67.35 | 73.81 | 74.07 | 74.45 | 74.83 |
| FitNet | 63.16 | 70.69 | 73.59 | 73.54 | 73.73 |
| AT | 58.58 | 71.84 | 71.73 | 72.73 | 73.32 |
| SP | 68.08 | 73.34 | 73.48 | 74.56 | 74.52 |
| CC | 65.43 | 70.25 | 71.14 | 71.29 | 71.38 |
| VID | 67.57 | 70.30 | 73.38 | 73.40 | 73.61 |
| RKD | 64.43 | 71.50 | 72.28 | 73.21 | 72.21 |
| PKT | 66.52 | 73.01 | 74.10 | 74.69 | 73.89 |
| AB | 67.20 | 70.65 | 73.55 | 74.31 | 73.34 |
| FT | 60.99 | 70.29 | 71.75 | 72.50 | 72.03 |
| NST | 64.96 | 71.28 | 74.12 | 74.68 | 74.89 |
| CRD | 69.11 | 74.30 | **75.11** | 75.65 | 76.05 |
| **SKD** | **69.26** | **74.41** | 75.08 | **76.02** | **76.42** |

Table 5: ImageNet experiments with ResNet18 student and ResNet34 teacher (Top1 accuracy)

| CE | KD | ES | SP | CC | CRD | AT | SRRL | SKD |
|---|---|---|---|---|---|---|---|---|
| 69.8 | 69.20 | 71.40 | 70.62 | 69.96 | 71.38 | 70.70 | 71.46 | **72.80** |

Table 6: Performance degradation problem on ImageNet.

| Teacher | Method | Accuracy | Teacher | Method | Accuracy |
|---|---|---|---|---|---|
| ResNet34 | KD | 69.43 | ResNet101 | KD | 68.91 |
| | ES | 70.98 | | **SKD** | **72.85** |
| | **SKD** | **72.80** | | | |
| ResNet50 | KD | 69.05 | ResNet152 | KD | 68.84 |
| | TA | 70.65 | | TA | 70.59 |
| | ES | 70.95 | | ES | 70.74 |
| | **SKD** | ***73.01*** | | **SKD** | **72.70** |

**ImageNet**  All experiments reported in Table 5 used ResNet34 as the teacher and ResNet18 as the student. Table 5 shows that SKD exceeds all of the previous SOTA by a large margin on ImageNet. Figure 1 shows the training process of vanilla KD and SKD. It is worth noting that SKD achieves comparable performance to KD's final performance after the first 30th epoch training.

### 3.2  PERFORMANCE DEGRADATION EXPERIMENTS

**CFIFAR-100**  We trained the ResNet14 with multiple teachers on the CIFAR-100 dataset. As shown in Table 7, the vanilla KD suffers from the performance degradation problem with oversized teachers (i.e., student accuracy continued decreasing when using teacher larger than ResNet32); while SKD continually improves the student performance as the teacher size is larger, which demonstrates that SKD can effectively alleviate the performance degradation problem. In addition, compared with vanilla KD, SKD significantly reduces both the entropy gap and free energy gap.

**ImageNet**  We compared SKD with two previous methods that aim to alleviate the degradation problem, Early Stop [4] (ES) and Teacher Assistant [20] (TA). Both of these two methods explicitly regularized the teacher capacity: 1) TA proposed to distill the large teacher to an intermediate teacher and then distill to the student, so that each knowledge distillation step has a better match between student and teacher capacity; 2) ES methods use the early stopped teacher, the teacher capacity would be regularized by fewer training steps. Table 6 shows the degradation problem in ImageNet

Table 7: Performance degradation problem on CIFAR-100. Student is ResNet14. SKD achieves lower training loss and higher accuracy. The entropy gap between the distilled student and teacher is also reduced significantly. Temperature is set to 4 in vanilla KD.

|  |  | ResNet20 | ResNet32 | ResNet44 | ResNet56 | ResNet110 |
|---|---|---|---|---|---|---|
| Training loss | Vanilla KD | 1.1 | 1.7 | 2.1 | 2.5 | 3.3 |
|  | SKD | **0.9** | **1.2** | **1.3** | **1.4** | **1.6** |
| Test acc | Vanilla KD | 67.4 | 68.2 | 68 | 67.5 | 67.1 |
|  | SKD | **68.2** | **68.7** | **68.9** | **68.8** | **69.2** |
| $G_{entropy}$ | Vanilla KD | 0.146 | 0.181 | 0.222 | 0.246 | 0.261 |
|  | SKD | **0.053** | **0.074** | **0.081** | **0.083** | **0.094** |
| $G_f$ | Vanilla KD | 0.042 | 0.053 | 0.063 | 0.068 | 0.074 |
|  | SKD | **0.029** | **0.042** | **0.041** | **0.045** | **0.044** |

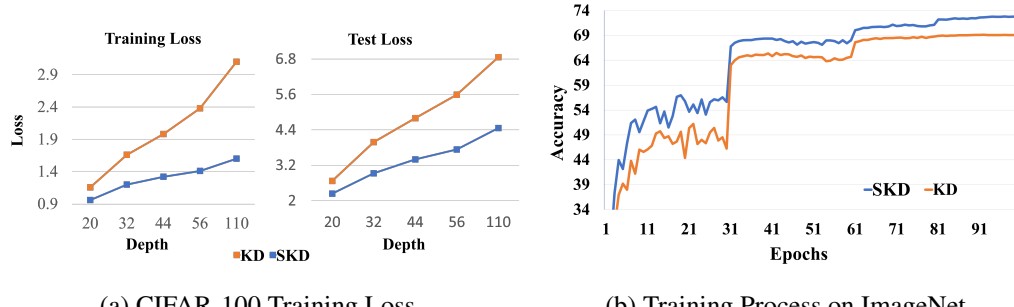

(a) CIFAR-100 Training Loss.  (b) Training Process on ImageNet.

Figure 1: (a) As the teacher size grows, the loss of SKD increases slower than KD, which shows that SKD alleviates the performance degradation Problem. (b) When trained on ImageNet, SKD achieves comparable accuracy with KD on ImageNet at the 30th epoch.

with ResNet18 as the student. We can see that SKD exceeds Early Stop and TA methods by a large margin in all teacher settings (Table 6). For example, when distilled by ResNet50 and ResNet152, the performance exceeded other methods by 2%. SKD achieves 73.01% accuracy, which is the best ResNet18 result that we know of.

## 4 RELATED WORK

Buciluǎ et al. [2] first proposed to compress a trained cumbersome model into a smaller model by matching the logits between them. Then Hinton et al. [11] advanced this idea and formed a more widely used framework known as knowledge distillation (KD). Knowledge distillation tries to minimize the KL divergence between the soft output probabilities generated by the logits through softmax function. Different from Xu et al. [35] that normalizes the features in the penultimate layer of the network to perform distillation, our methods perform normalization on the logits layer. Furthermore, knowledge distillation can also be regarded as a soft label training method. Specifically, previous studies have found that knowledge distillation helps to regularize the training of network. The relationship between KD and other regularization techniques (e.g., label smoothing) has been discussed in various works [21, 29].

Although distillation has shown a great potential in many tasks, researchers found that larger teachers often unexpectedly harm the distillation performance, despite their more powerful ability [4, 20]. The performance degradation problem is particularly severe on ImageNet, resulting in poor performance of distilled student model. It was widely accepted that the capacity mismatch between teacher and student causes this problem [40]. Previous research proposed to regularize the teacher capacity to alleviate this problem heuristically. For example, Cho & Hariharan [4] proposed to early stop the training of the teacher. Moreover, Mirzadeh et al. [20] proposed to use a medium-size teacher assistant (TA) to perform a sort of sequence distillation.

## 5 CONCLUSION

The vanilla knowledge distillation overlooks the entropy gap between the student and the teacher, which may cause the performance degradation with oversized teachers. We presents the Spherical Knowledge Distillation framework, which address the performance degradation problem by reducing the gap between the teacher and student. We validate the effectiveness of our method on CIFAR-100 and ImageNet. Experimental results show that SKD can effectively mitigate the performance degradation problem and produce competitive students.

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
