# OpenReview forum: "AN ENTROPY PERSPECTIVE IN KNOWLEDGE DISTILLATION"
_ICLR.cc/2024/Conference — ICLR 2024 Conference Withdrawn Submission_

### Official Review · Reviewer_kj6Z · 2023-10-20

**Soundness:** 2 fair
**Presentation:** 2 fair
**Contribution:** 2 fair
**Rating:** 3
**Confidence:** 4

**Summary:**

This paper analyzed reasons of performance degradation of distilling knowledge from an overly large teacher model to the student, and proposed to narrow entropy gap and free energy gap between the student model and the teacher model.  Through theoretical analysis, the paper proposed an alignment  module to narrow these gaps, thus enhancing the student performance.  Experiment results validated the effectiveness of the proposed alignment module.

**Strengths:**

1. The paper provided theoretical analysis. And, these mathematical proof is easy to understand
2. The author's writing is very good, and the entire paper is relatively easy to understand

**Weaknesses:**

1. The contribution of this method is not very important. In section 4, the authors summarized that "Different from Xu et al. [35] that normalizes the features in the penultimate layer of the network to perform distillation, our methods perform normalization on the logits layer.".  Although the paper is different from Xu et al. [35], it is not difficult to derive this method from Xu et al. [35].
2. The paper proposed two metrics (entropy gap and free energy gap) to evaluate the gap of the student and the teacher. They are very close as shown in Eq. 9 and Eq.13, so they cannot provide more information from different perspectives.
3. Some important works are missing and not to compare, e.g. Zhao, Borui, Quan Cui, Renjie Song, Yiyu Qiu and Jiajun Liang. “Decoupled Knowledge Distillation. 2022 IEEE/CVF Conference on Computer Vision and Pattern Recognition (CVPR) (2022): 11943-11952.
4. || * || is not exactly indicated, it is better to add explanations, e.g. L1 or L2, or others

**Questions:**

1. When proving equation 8, the Taylor expansion is a first-order expansion, and when proving equation 12, the Taylor expansion is a second-order expansion. Why use different formulas?
2. There are no experiments showing the combination of SKD with other KD methods. SKD adds an alignment module, so it can be directly combined with other KD methods. Can it improve the performance of other KD methods?
3. In section 3, there are duplicate descriptions ", e.g. Learning rate is dropped at 30, 60, 80 and 90 epoch." and "learning rate dropped by 0.1 in 30, 60, 80, 90 epoch".

---

### Official Review · Reviewer_YbW2 · 2023-11-02

**Soundness:** 3 good
**Presentation:** 3 good
**Contribution:** 2 fair
**Rating:** 3
**Confidence:** 4

**Summary:**

This paper investigates knowledge from the entropy perspective and proposes a knowledge distillation method by first normalizing the logits and then minimizing the KL-divergence. In this way, the entropy gap can be reduced. Experiments on CIFAR-100 and ImageNet are conducted and show promising results.

**Strengths:**

1. This paper is easy to follow.
2. The proposed method is easy to implement.
3. Comparison with different methods are reported.

**Weaknesses:**

1. The improvement is minor. As shown in Table 4, there is almost no improvements over CRD (ICLR'2020).
2. The proposed method of normalizing the logits is not a big contribution, since normalizing the logits is widely used in metric learning and few-shot learning literature.

**Questions:**

1. First, this paper defines entropy gap, which I think is reasonable. Then, this paper simply assumes a small entropy gap leads to a better performance and proposes a method to minimize the entropy gap. However, there is no theory or comprehensive experiments supporting this assumption. When this assumption works, and for what kinds of teacher and student networks?
2. CRD was proposed in 2020. Recent SOTA approaches are not compared.

---

### Official Review · Reviewer_dqtM · 2023-11-03

**Soundness:** 2 fair
**Presentation:** 2 fair
**Contribution:** 2 fair
**Rating:** 3
**Confidence:** 2

**Summary:**

This paper focuses on the performance degradation of student models when the teacher model is significantly large. The authors provide theoretical results on the discrepancy of student and teacher models, in terms of entropy rates & free energy. The authors also suggested spherical knowledge distillation, which is shown to perform well on multiple benchmarks.

**Strengths:**

* The authors tested their scheme for multiple settings.
* The empirical results are promising for some cases.

**Weaknesses:**

* The theoretical results are quite easy to derive. From the definition, entropy and free energy is bit related, so figuring out the relationship in the distillation setup does not seem to be a critical contribution.
* It is unclear why simple normalization of logics (as in eq.15) is motivated by the theoretical results. The authors say “It can be seen that ||v||^2 - ||z||^2 plays a key role and can be removed by normalization techniques”, but why are we removing the important term having key role?
* I personally believe current theoretical explanation is not enough to justify why spherical KD is working much better than vanilla KD.

**Questions:**

* It’s a minor issue, but the title in the manuscript is different from the title in the console.

---

### Public Comment · ~Peter_Chen5 · 2023-11-20
**variance of the results**

Dear author, is it possible to release the official implementation of the paper, after multiple trials, we failed to reproduce your results, with high variance.
Best